# Kinetic trapping organizes actin filaments within liquid-like protein droplets

Aravind Chandrasekaran[1], Kristin Graham [2], Jeanne C. Stachowiak [2,3] ✉ & Padmini Rangamani [1] ✉

Several actin-binding proteins (ABPs) phase separate to form condensates capable of curating the actin network shapes. Here, we use computational modeling to understand the principles of actin network organization within VASP condensate droplets. Our simulations reveal that the different actin shapes, namely shells, rings, and mixture states are highly dependent on the kinetics of VASP-actin interactions, suggesting that they arise from kinetic trapping. Specifically, we show that reducing the residence time of VASP on actin filaments reduces degree of bundling, thereby promoting assembly of shells rather than rings. We validate the model predictions experimentally using a VASP-mutant with decreased bundling capability. Finally, we investigate the ring opening within deformed droplets and found that the sphere-to-ellipsoid transition is favored under a wide range of filament lengths while the ellipsoid-to-rod transition is only permitted when filaments have a specific range of lengths. Our findings highlight key mechanisms of actin organization within phase-separated ABPs.

Phase separation has been implicated in locally concentrating many proteins that are involved in signal transduction[1,2] and in the remodeling of the actin cytoskeleton[3,4]. While the conditions under which droplets form have gained significant interest recently[5–8], studies of the downstream kinetic consequences due to phase separation have been limited to transcriptional condensates[7,9–11]. Significant progress has been made in our understanding of the formation and dynamics of cytoskeletal condensates[12–14]. Recently, we showed that Vasodilator Stimulated Phosphoprotein (VASP), a processive actin polymerase and an actin-bundling protein, forms liquid-like droplets in vitro under physiological conditions[15]. When actin was added to these VASP droplets, we found that the actin filaments self-organized into different droplet-encapsulated structures such as shells, rings, or disks depending on the relative concentrations of actin and VASP. Particularly, when the actin ring thickness exceeded a critical value, the actin filament bundle was able to deform the droplet into an ellipsoidal disk or open up to form a linear actin bundle within a rod-like droplet. Arguments from an energy minimization standpoint[15] identified the

critical ring thickness above which a spherical droplet will deform into an ellipsoid. However, a critical transition prior to the elongation of the droplet is the formation of ring-like structures of actin. To broaden our understanding of droplet-driven kinetic assembly, here, we study the biophysical mechanisms that give rise to different actin configurations within a VASP droplet.

The interactions of actin with VASP have been studied previously both in cellular[16,17] and reconstituted systems[18,19]. VASP was originally isolated from platelets[20] where it is known to inhibit platelet activation/aggregation[21,22]. When platelets are exposed to vasodilators, cAMP and cGMP levels are elevated, resulting in the kinase-mediated activation of VASP[23,24]. VASP has been subsequently found in several other cell types[25] and has been implicated in cellular motility[26] and axonal guidance[27–29]. Subsequent discovery of Ena and Ena-VASP-like (EVL) proteins resulted in a new Ena/VASP family of proteins. The Ena/VASP family of proteins is also of importance clinically due to its role in promoting Listeria motility[30,31] and cancer cell metastasis[32–34]. Ena/VASP proteins are characterized by three major domains: Ena-VASP

[1]Department of Mechanical and Aerospace Engineering, University of California San Diego, La Jolla, CA 92093-0411, USA. [2]Department of Biomedical Engineering, University of Texas at Austin, Austin, TX 78712, USA. [3]Department of Chemical Engineering, University of Texas at Austin, Austin, TX 78712, USA. ✉e-mail: jcstach@austin.utexas.edu; prangamani@ucsd.edu

homology domains (EVH) 1 and 2, which are separated by a proline-rich region. The EVH1 domain contains binding sites for focal adhesion complexes such as Zyxin and Vinculin[31]. The EVH2 domain contains the coiled-coil tetramerization domain[35–37], along with G-actin and F-actin-binding motifs[38,39]. Therefore, at the molecular level, VASP tetramers act as weak crosslinkers[40] and processive actin polymerases[18,41]. While the role of Ena/VASP proteins in controlling the elongation of actin filaments has been well established, the emergent mechanism of network organization from Ena/VASP crosslinking is not yet fully understood.

Actin filaments form multiple architectures depending on their binding partners and the nature of confinement[42–44]. Crosslinked shell and ring-shaped networks are of biological relevance, particularly in actin cortex[45,46] and cytokinetic ring[47], respectively. Actin cortex dynamics has been extensively studied to understand the role that cortical tension gradients[48,49] play in determining cell shape[50,51], cell migration[52,53], and in the formation of stress fibers[54]. Additionally, cytoskeletal ring formation and the subsequent contractility[55] enable cytokinesis in fungal and metazoan cells[56–58]. Reconstituted actin networks within oil-in-droplet emulsions and giant unilamellar vesicles (GUVs) have identified a minimal set of components and also offer mechanistic insights towards the formation of shells[59–61] and rings[62,63]. Broadly, non-interacting filaments form shells, while adequately crosslinked filaments form rings. Additional mechanisms that aid ring formation are membrane-filament anchors and actin-bending proteins such as septins[64,65]. The nature of confinement of actin filaments also plays a critical role in determining actin network shape[66]. Non-interacting filaments within liposomes or oil-in-water drops, peripherally accumulated to form actin shells. On the other hand, actin networks within microfabricated shallow chambers without any boundary interactions undergo bundling and peripheral accumulation. Such transitions are not found in bulk where isotropic-to-nematic transitions are preferred[67]. Exploring the role of multiple crosslinkers, ref. 68 showed that fascin and actin sort spatially along aster-shaped networks within GUVs and crosslinker ratio affects GUV shape[69]. However, the role of dynamic crosslinking in controlling bundling and network shape has not been fully understood.

Computational efforts have played a key role in dissecting the biophysics of these non-equilibrium processes[70–73], particularly in the formation of shell- and ring-shaped networks under confinement. Simulations of rigid droplets with self-avoiding filaments are shown to form shells with a nematic surface configuration that depends on contour length[74] and bending modulus[75]. Network organization of microtubules with motor proteins ranges from symmetric asters to cortical bundles as the confining radius is increased[76]. Ni et al. [77] show that actin shells form when treadmilling filaments are confined within spherical volumes (filament length <sphere radius) with explicit crosslinkers and motors. This study does not explore the role of multivalent crosslinkers. Exploring the role of filament lengths along with a filament-filament attraction potential (implicit, permanent crosslinking), ref. 72 demonstrate that growing filaments of lengths comparable to or greater than confinement diameter form rings under appropriate inter-filament attraction parameters, and form shells otherwise. Consistent with this expectation, ring probability increases with filament-boundary attraction[62,72]. Additionally, they show that filaments within prolate ellipsoids align along the axis of lowest curvature consistent with our previous energetic framework[15]. Owing to the implicit crosslinking model, these studies do not consider the role of steric repulsion between crosslinkers and the role of diffusive, reactive time scales of crosslinking, both of which are crucial for determining chemical propensities under crowded environments[78,79]. Additionally, the role of network remodeling and competition between surface and filament energy in altering the shape of the boundary remains unexplored in the presence of multivalent crosslinkers. In this work, we sought to investigate the key biophysical determinants of such varied structures using a combination of computational modeling and experiments. Specifically, we hypothesized that different actin networks could be characterized by kinetic accessibility or lack thereof resulting from actin-VASP interactions.

We used Cytosim, an agent-based modeling framework[71,80–82], to model dynamics of the actin filaments confined within an actin droplet at a non-equilibrium steady state and investigated how different kinetic time scales, including binding and unbinding of actin to VASP, and actin elongation rate affect the formation of actin shells and rings. Our simulations showed that the distribution of rings and shells is determined by the competition between binding and unbinding kinetics of actin and VASP. Other time scales, such as actin elongation rate, do not significantly alter ring formation. Rings form when the initial bundling is effective with a longer residence time for VASP to crosslink filaments (specific binding and unbinding rates), otherwise shells form. Our predictions of effective actin bundling as a determinant of shells versus rings were validated using a mutant of VASP that has decreased bundling capability and effectively lowered residence time. Finally, the shape transition of VASP droplets from spheres to ellipsoids to rods, driven by ring formation, is determined by the maximum filament length. Thus, our study identifies how different kinetic parameters can determine the actin network architecture and consequently VASP droplet deformation. From a biological perspective, this work identifies specific physical mechanisms by which protein condensates could influence the architecture of the cytoskeleton.

## Results

We begin by constructing a LAMMPS[83] model of a VASP droplet inspired by the method outlined by ref. 84. We approximate VASP and crowder molecules as spherical particles. Considering attractive Lennard-Jones (LJ) potentials between VASP molecules and truncated-repulsive LJ potentials between VASP and crowder molecules, we reproduce a phase-separated droplet in Supplementary Fig. 1A, B. We represent actin filaments as a string of LJ beads[85–87]. Please refer to Supplementary Methods and Supplementary Table 1 for a detailed description of the methods and model parameters used. When actin filaments are equilibrated within the droplet and are allowed to be permanently crosslinked (Supplementary Fig. 1C, D), we see that the filaments stay within the droplet and also lead to deformation of the droplet (Supplementary Fig. 1 E, F). These results are consistent with our continuum model[15]. Since it is computationally expensive to explore all biologically relevant parameters (such as crosslinker kinetics and filament growth rate) using the LAMMPS model, we consider the following approximations. We construct a minimal computational model to study how the condensates affect actin organization within droplets. The model incorporates the dynamics of actin filament elongation and VASP-driven bundling. Specifically, we sought to identify the conditions in which actin can form shells and rings within VASP droplets, as observed in experiments (Fig. 1A)[15]. We assume that the VASP droplets have already formed and are characterized by a crowded environment with a high density of VASP tetramers. Experiments showed that polymerized actin remains confined inside the droplet[15]. We consider actin networks inside droplets under a high surface tension limit using non-deformable droplet-mimics characterized by a hard wall boundary condition. We employ a quasi-static perturbation of droplet shape later in the paper to understand how it affects actin organization. Additionally, we resolve just a subset of the VASP tetramer molecules within the droplet. Based on these assumptions, we simulated VASP droplets using CytoSim[80,81], an agent-based simulation platform designed to simulate cytoskeletal networks. In CytoSim, actin filaments are represented as a series of cylinders, each of length $L_{seg}$ and with a flexural rigidity determined by the persistence length of actin filaments[88]. Actin filaments are allowed to elongate at a predetermined rate (Fig. 1C). VASP tetramers are modeled as multivalent linkers[73]. Each of the four binding sites on the VASP tetramer can

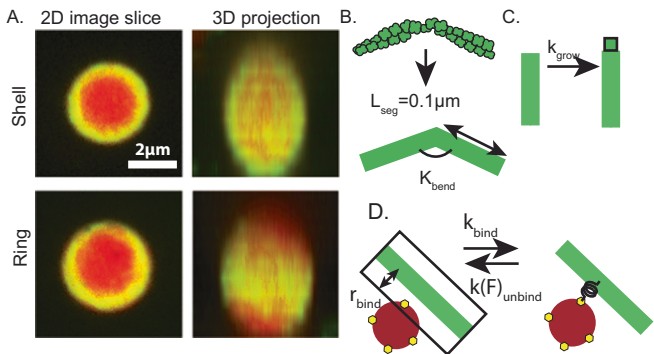

**Fig. 1 | Computational model to study actin organization in VASP droplets.**
**A** 2D confocal image slice (representative image from $n = 3$ replicates), and corresponding 3D reconstructions from experimental images showing a VASP droplet with shell- and ring-shaped actin networks from experiments. VASP is in red and actin is in green. Images show the merge of both channels. Scale bar 2 µm.
**B**–**D** Simulation set up in Cytosim for actin filaments. Please refer to Supplementary Table 2 for all relevant model parameters. The VASP droplet is represented as a spherical reaction volume with hard wall boundary conditions to mimic the high surface tension limit. **B** The actin double helix is represented as a series of line segments, each of length $L_{seg}$ and with a flexural rigidity determined by the persistence length of actin filaments[88]. **C** Filaments are allowed to grow deterministically with extension rate $k_{grow}$. **D** VASP tetramers are represented as spherical crosslinkers with four binding sites. When a binding site encounters a filament segment within the binding distance $r_{bind}$, a stochastic binding reaction occurs with probability determined by the mesoscopic binding rate $k_{bind}$. Additionally, we consider slip bond kinetics in accordance with Bell's law. All parameters are given in the supplementary material.

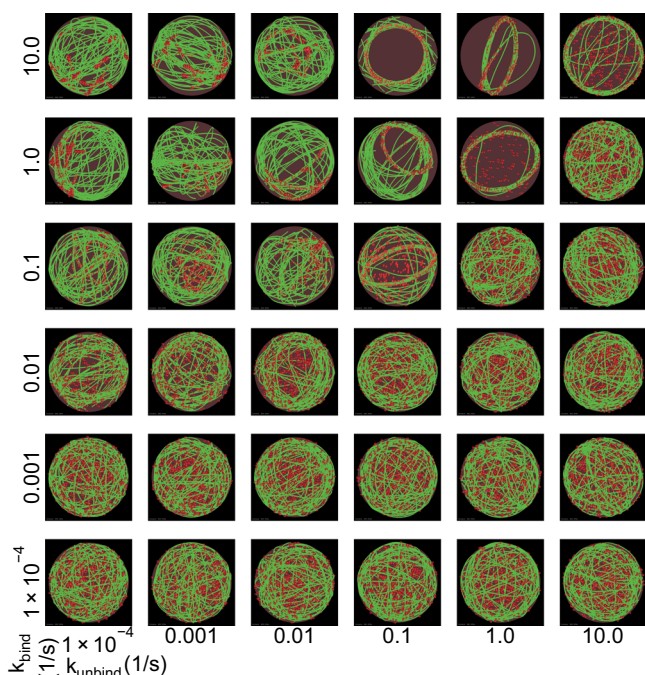

**Fig. 2 | Varying crosslinker kinetics generates experimentally observed actin network shapes.** Representative snapshots at $t = 600$ s from simulations with 30 seed actin filaments are shown, corresponding to different crosslinker binding rates ($k_{bind}$, varied along each column) and unbinding ($k_{unbind}$, varied along each row). Actin filaments are shown in green, while the tetrameric crosslinker is shown as red spheres; also see Supplementary Movie M1.

stochastically bind actin filaments that are within a binding distance of 30 nm, depending on the specified mesoscopic binding rate, $k_{bind}$ (Fig. 1D). Actin-VASP unbinding is modeled as a force-sensitive rate, given by $k_{unbind}$. Given the stochastic nature of the simulations, we conducted multiple replicates per condition. Using this simulation platform, we systematically investigate how different biochemical and mechanical parameters can impact the architecture of the actin network in VASP droplets.

## The binding-unbinding kinetics between actin and VASP determines actin network organization in VASP droplets

We begin by simulating a spherical volume ($R_{drop} = 1$ µm) with a tetrameric crosslinker concentration of 0.40 µM. In this volume, 30 seed actin filaments, each of length 0.1 µm, are distributed at random orientations throughout. The filament elongation rate is set to 10.3 nm/s such that at 600 s, the final length of actin filaments is the cross-section circumference of the droplet, $2\pi$ µm (final actin concentration is 27.67 µM). The simulation time of 600 s was chosen based on experimental observations[15]. With these initial conditions, we varied the binding and unbinding rate of VASP to actin. The dissociation constant of VASP-actin was measured experimentally to be 1.8 nM[18]. However, how this dissociation constant alters the dynamic evolution of the actin network in VASP droplets, particularly due to the multivalent nature of VASP tetramers binding to actin filaments, remains unclear. Therefore, we varied $k_{bind}$ and $k_{unbind}$ from $1 \times 10^{-4}$ s$^{-1}$ to 10 s$^{-1}$ in our simulations (Fig. 2) for a fixed concentration of actin and tetramers.

Our simulations reveal that when $k_{bind}$ (Fig. 2 columns) and $k_{unbind}$ (Fig. 2 rows) are varied, the resulting actin networks resemble experimentally observed networks such as shells and rings, in addition to networks that share features of rings and shells (Fig. 2). This result predicts that the crosslinking time scale determines the network organization of confined actin networks. For the combination of parameters investigated, the formation of shell-like structures is more likely. The formation of VASP-decorated actin rings is favored when

$k_{bind}$ is high (1 and 10 s$^{-1}$) and $k_{unbind}$ is intermediate (1/s). Specifically, if $k_{unbind}$ is too high, then actin filaments will not be sufficiently bundled to form a ring. In contrast, if $k_{unbind}$ is too low, then the initial, unaligned contacts will be unable to rearrange into an aligned bundle configuration. Thus the actin shape accessibility is controlled by the time scale of actin-VASP interaction. Additionally, we note that the plane in which the ring is oriented is random in the simulations but determines the plane in which the droplet deforms, as we will show later (see Fig. 7)[15].

Next, to quantify the actin architecture under these different conditions, we calculated the fraction of spherical surface occupied by actin as described in Methods (Fig. 3A, B, Supplementary Fig. 2A–C). At the final time point, we expect actin rings to occupy a lower surface area fraction of the boundary and shells to occupy a higher surface area fraction of the boundary (Fig. 3A). Calculating the evolution of area fraction over time, we found that for all combinations of parameters tested, the initial dynamics are fairly similar for all conditions for up to 200 s. This is understood by recognizing that at $t > 184.47$ s, the filament reaches length $>2R_{drop}$ and then starts to bend (Supplementary Fig. 3). At low binding rates ($\leq 0.001$/s), for all values of unbinding rates tested, we found that the surface area fraction continued to grow with time, consistent with the shell-like actin organization in Fig. 2. At higher binding rates, we found that for certain values of unbinding rates (for example, $k_{bind} = 1$ s$^{-1}$, $k_{unbind} = 10$ s$^{-1}$), the actin network formed shells at later time, while other area fraction trajectories seemed to stabilize to a smaller area fraction value (for example, $k_{bind} = 1$ s$^{-1}$. $k_{unbind} = 1$ s$^{-1}$). Such trajectories can be mapped to ring-like actin networks. We also observed that a few trajectories appeared to settle between shells and rings, suggesting that other intermediate structures appear in the kinetic space sampled. The strong dependency of these trajectories on kinetic rates suggests that the architectures of actin networks within VASP droplets represent kinetically trapped states.

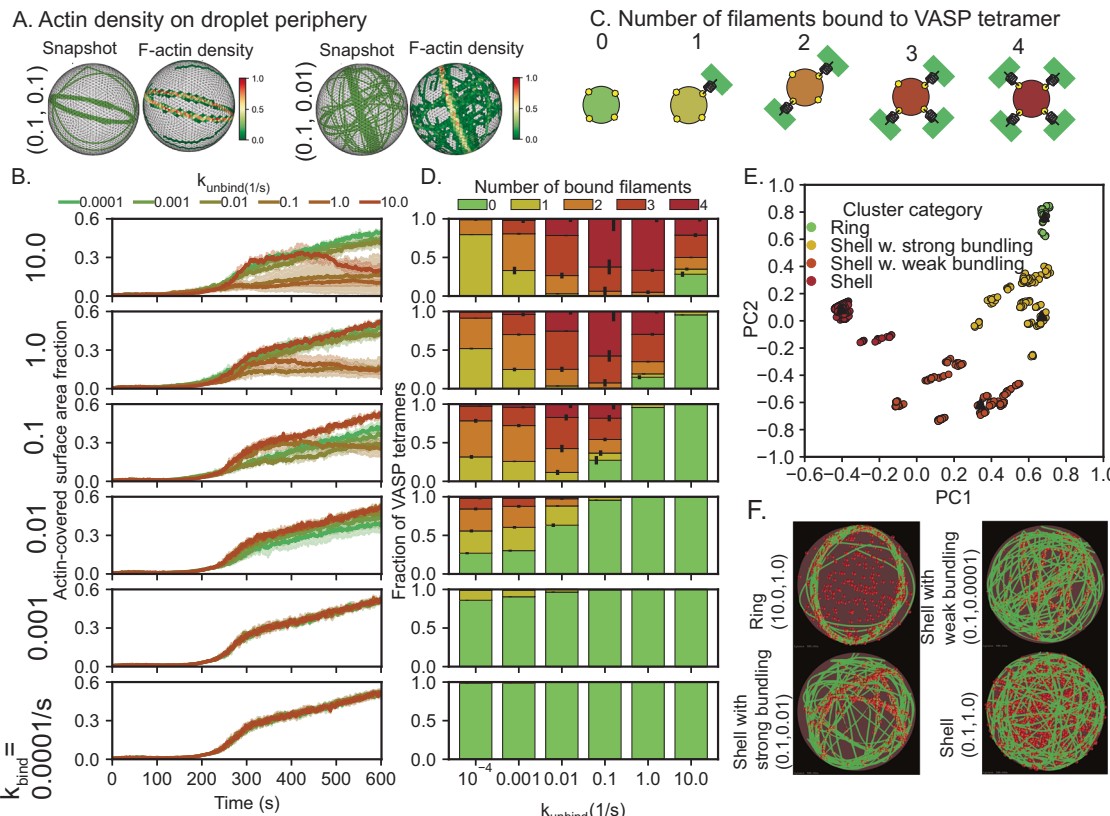

**Fig. 3 | Signatures of different actin network configurations. A** Representative final snapshots corresponding to two different kinetic parameters (given as ($k_{bind}$, $k_{unbind}$) to the left) are shown along with the surface actin density along the icosphere. The triangulated surface is colored based on local actin abundance (normalized). **B** Each panel shows the kinetics of the surface area fraction of the sphere covered in F-actin. The binding rate is changed across panels while each panel shows the role of VASP unbinding rate. (three replicates per condition) **C** Cartoon illustrating the multivalent nature of VASP tetramers. **D** Each panel shows a stacked bar graph illustrating the fraction of VASP tetramers bound to 0, 1, 2, 3, and 4 filaments for different VASP unbinding rates. Error bars show standard deviation.

The corresponding VASP binding rate is shown to the left of panel B and the error bar shows the standard deviation. For (**B**, **D**), data were obtained from the last 5% of each of the five replicate trajectories (30 snapshots). **E** Five snapshots per replicate were considered from each of the 36 kinetic parameters studied in Fig. 1, and the data shown in panels **B** and **D** were used to cluster data using K-means clustering. The resulting four clusters using the first two principal components (PCs) are shown in (**F**). Along with the snapshot (identified by the $k_{bind}$, $k_{unbind}$ pair) closest to each cluster centroid (solid triangles). Source data are provided as a Source Data file.

To further classify these networks, we calculated the extent of filament bundling (see Supplementary Material for details). The extent of bundling was quantified by counting the number of crosslinkers that are bound to 0 (unbound), 1, 2, 3, or 4 filaments (Fig. 3C, D). We found that conditions that formed shell networks are associated with poor bundling and have a high number of unbound crosslinkers (Fig. 3D, left column) because the residence time of crosslinkers is very low. On the other hand, ring networks have a higher fraction of crosslinkers bound to at least three filaments (Fig. 3D right column). Finally, shell-ring intermediates have a significant number of crosslinkers bound to more than two filaments.

Thus far, our analysis has been based on visual inspection of the actin networks and corresponding metrics (area fraction and non-zero number of filaments bound to a crosslinker, a total of five order parameters). These five dimensions are not independent but correlated to some extent (Supplementary Fig. 4). We reduced the dimensionality of the system using principal component analysis and found that the first three principal components (PCs)[89] explained 97.64% of the variance of the entire dataset (Supplementary Fig. 4A). To interpret the relative contributions of the original five dimensions on each of the three PCs, we employed varimax rotation to calculate the loadings. We found that each of our five order parameters contributes strongly to at least one of the three PCs. Additionally, we observe from the varimax loading (Supplementary Fig. 4B) that PC1 is positively

correlated with the number of crosslinkers bound to three and four filaments. Additionally, it is negatively correlated with surface area fraction. Plotting the first two PCs, we see that shells have a low coordinate along PC1 consistent with both a high surface area fraction, and the low number of VASP molecules bound to three and four filaments. Therefore, we hypothesized that the various actin shapes in our dataset should be localized to distinct regions of the PC space. Employing K-means clustering analysis along the first three PCs, we identify four distinct clusters in our dataset. (Fig. 3E, F and Supplementary Fig. 4C). Visual inspection of snapshots within each cluster reveals that shells and rings form two distinct clusters, while the other two clusters consist of shells with weak bundling and shells with strong bundling (Fig. 3E, F). To understand the shape differences between the clusters, we computed the radius of the gyration tensor for each of the actin shapes and calculated the spans approximating the network shape as an ellipsoid. The distribution of ellipsoidal spans are plotted for each of the clusters in Supplementary Fig. 5, and we see distinct differences in shapes between each of the four clusters identified. Grouping the snapshots based on their $k_{bind}$, $k_{unbind}$ values, we find that in a subset of conditions, the emergent actin shape is probabilistic (Supplementary Table 3). These findings predict that a nontrivial combination of actin-VASP binding-unbinding rates and the number of filaments bound to each VASP molecule determine the network architecture.

## Actin ring formation is robust to changes in filament elongation rate

We next asked if the kinetics of actin elongation could affect the probability of ring formation. We conducted two separate sets of simulations to answer this question—we varied the final filament length (Fig. 4) and the actin elongation rate (Fig. 5). These two variations were designed based on the knowledge that VASP promotes actin filament elongation as a processive polymerase and the polymerization rates can depend on the actin organization itself. VASP is known to gather and promote the elongation of multiple barbed ends simultaneously, and VASP-mediated polymerization rates of shared barbed ends can be up to three times faster than free barbed ends[18].

We fix the $k_{bind}$, $k_{unbind}$ values where we know rings will form ($10\,s^{-1}$ and $1\,s^{-1}$, respectively) and change the maximum filament length, $L^{max}_{fil}$ that can be attained in the simulations at 600 s (Fig. 4). We explore the role of droplet diameter and circumference in controlling actin shape by simulating $L^{max}_{fil} < R_{drop}$, $R_{drop} < L^{max}_{fil} < 2R_{drop}$, $2R_{drop} < L^{max}_{fil} < \pi R_{drop}$, and $\pi R_{drop} < L^{max}_{fil} < 2\pi R_{drop}$. The number of filaments and actin addition rate were chosen such that the rate of F-actin addition remains constant. Thus, in this simulation, we are effectively changing the elongation rate of individual filaments by controlling the final filament length. For all filament lengths, we see that actin forms bundles, consistent with the role played by the binding and unbinding rates. When the maximum filament length is smaller than the radius of the droplet, we see that the bundles are linear (Fig. 4A i-ii) and probability distribution of the end-of-end distances of filaments peaks at distances smaller than the droplet diameter (Fig. 4B i). When the filament length is greater than the radius of the droplet, but less than the diameter of the droplet, we observe bent bundles (Fig. 4A iii-iv) and actin distribution shifts closer to the diameter but not quite towards rings (Fig. 4B i). When the filament length is greater than the diameter of the droplet, we observe that the filament bundles always form rings (Fig. 4A, v-viii), and the end-to-end distance probability is close to the droplet diameter (Fig. 4B ii). The peak observed close to droplet diameter in ring networks stems from the fact that the filament ends are

uniformly distributed along the circumference of the ring. These results are consistent with findings from computational studies with implicit crosslinkers[72].

We also studied the role of actin addition through actin nucleation (Supplementary Fig. 6A). We simulated the effect of linear nucleator concentration whose kinetic parameters are similar to Arp2/3. Please refer to Supplementary Table 4 for a detailed description of the model parameters used. The total actin concentration is fixed and the growth rate is proportional to the amount of unpolymerized actin. Enhanced nucleation (Supplementary Fig. 6B) at high nucleator concentration reduces median filament length (Supplementary Fig. 6C). Further, we see that the network is made of a population of short and long filaments. The wider distribution of filament lengths leads to a well-bundled network with most of VASP tetramers bound to 3+ filaments (Supplementary Fig. 6D). While the concentration of the nucleator is less than 50 nM, we see an increase in actin-covered surface area (Supplementary Fig. 6E) suggesting that the droplet periphery acts as the zone of VASP-mediated bundling consistent with representative images shown in Supplementary Fig. 6A and the radial density profile shown in Supplementary Fig. 6F. When the nucleator concentration exceeds 50 nM, we see that the maximum filament length drops below $\pi R_{drop}$ (Supplementary Fig. 6C). Thus, we find that VASP-mediated bundling of these networks results in enhanced actin concentration at the droplet core (Supplementary Fig. 6A, F).

Next, we fixed the final filament length and varied the filament elongation rate. We define the time it takes a filament to grow to length $2\pi R_{drop}$ (one circumference along the droplet) as $T_{2\pi R}$. We vary $T_{2\pi R}$ between 10 s and 600 s (Fig. 5). Once the filaments elongate to $2\pi R_{drop}$, any further filament elongation is not permitted. To understand the role of $T_{2\pi R}$, we chose binding parameters where ring formation was probabilistic ($k_{bind} = 0.1/s$, $k_{unbind} = 0.1/s$). We also varied the VASP concentration to investigate if it had a role to play in ring formation. We observed that actin rings can be obtained for a number of combinations of elongation rate and VASP concentration (Fig. 5A). Furthermore, we observed that for a fixed VASP concentration (Fig. 5B,

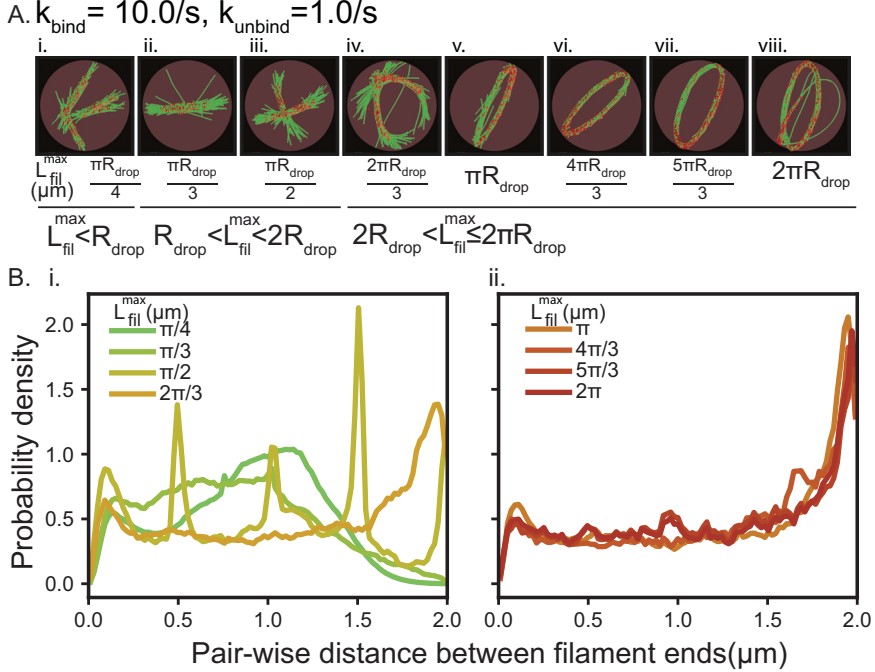

**Fig. 4 | Actin networks with growing filaments form rings when the final filament length is greater than the diameter of the droplet. A** Representative final snapshot ($t = 600$ s) from simulations under ring-forming conditions at various final filament lengths; also see Supplementary Movie M2. **B** Probability density functions

of pairwise distances between filament ends is shown under various $L_{fil}^{max}$. (Data used: Final five snapshots from each of the five replicates per condition). Source data are provided as a Source Data file.

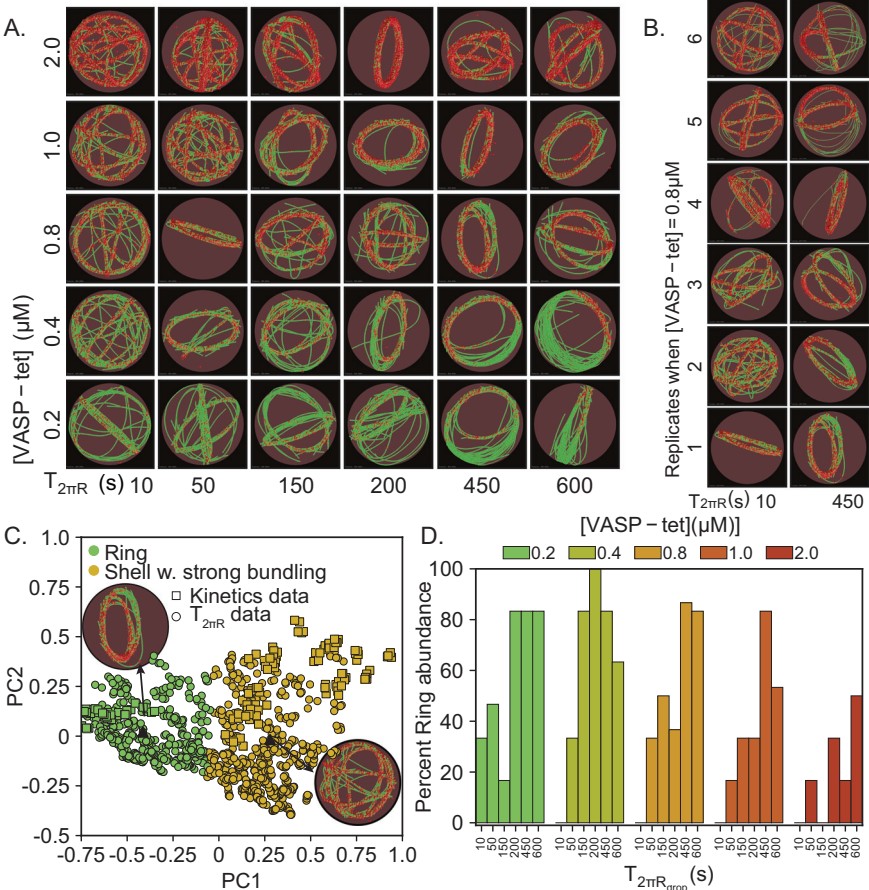

**Fig. 5 | Simulations predict slower elongation rate (high $T_{2\pi R}$) increases ring formation probability. A** Representative final snapshots ($t = 600$ s) from simulations at various [VASP-tet] and elongation rates measured by $T_{2\pi R}$, the time for filaments to grow to $2\pi R_{drop}$ length ($k_{bind} = 1.0$/s, $k_{unbind} = 0.1$/s); also see Supplementary Movie M3. Rings are more probable when $T_{2\pi R}$ is large. **B** We find that the final network shape has higher heterogeneity at low $T_{2\pi R}$ (data shown for tetramer concentration of 0.8 μM from six replicates, rest shown in Supplementary Fig. 7). **C** Simulation results from this study (circle) were combined with data from Fig. 3E (shown as squares) and clustered to identify salient network shapes. Combined

data projected along the first two principal components are shown, colored by cluster. The data points from the current study that are closest to each of the cluster centroids are shown as triangles, along with insets of the final snapshot. Supplementary Fig. 9 shows details of the clustering algorithm and Supplementary Fig. 10 shows the distribution of ellipsoidal span for rings and rings with strong bundling **D** Probability of ring formation calculated from the clustered data is shown as a function of [VASP] and filament elongation time scale. Data used: Five snapshots from each of the ten replicates. Source data are provided as a Source Data file.

showing VASP = 0.8 μM, see Supplementary Fig. 7 for others), the network configuration was more heterogeneous across replicates at lower values of $T_{2\pi R}$ (faster elongation) than at higher values of $T_{2\pi R}$ (slower elongation). To quantify the variation in the actin networks, we plotted the principal components by combining the earlier dataset (Fig. 3E) with the current dataset. Please refer to Supplementary Fig. 8 for plots showing overlay of data from this study with that shown in Figure and Supplementary Fig. 7 for images of replicates at other [VASP-tet]. We found that despite the apparent differences in the network architecture, only two possible network configurations were seen−actin rings and shells with strong bundling (Supplementary Fig. 8). We repeated the clustering and principal component analyses using an updated dataset with data points corresponding to the ring and shells with strong bundling from the earlier dataset and all data points of the current dataset (Fig. 5C). Finally, we found that while the ring probability increases with slower elongation rate (larger $T_{2\pi R}$), the remainder of the probability is to form shells with strong bundling (Fig. 5D). Thus, we find that actin ring formation probability is robust to changes in actin elongation rate as long as the $k_{bind}$ and $k_{unbind}$ are favorable for ring formation, suggesting that actin-VASP interaction time scale dominates ring formation.

To understand the limits of actin concentration that will lead to ring formation, we studied networks with an increasing number of

actin filaments, with $T_{2\pi R} = 600$ s (Supplementary Fig. 11A). We classify the resulting trajectories by combining them with the data points corresponding to rings and shells with strong bundling from Fig. 3E. Supplementary Fig. 11B shows details of principal components used and that the dataset has three unique clusters corresponding to the shell with strong bundling, ring/multi-ring networks, and networks where bundling is restricted to sub-sections of the network while the rest of the network remains diffuse (Supplementary Fig. 11C). Plotting the probability of finding networks in each cluster (Supplementary Fig. 11D), we find that increasing actin leads to abundance in locally-bundled diffuse networks inversely proportional to [VASP-tet].

**Weakening filament bundling results in the loss of rings**

Thus far, our simulations predict that ring formation occurs when the initial bundling is effective (large effective residence time for actin-VASP multivalent binding), and that ring formation is robust to changes in actin elongation rates. If this is true, then decreasing the residence time of VASP tetramers on actin filaments should change the kinetic trajectories from rings to other network configurations. To test this hypothesis, we choose a $k_{bind}$, $k_{unbind}$ pair where rings form (1, 1) and slowly increase $k_{unbind}$ (Fig. 6A). Increasing $k_{unbind}$ decreases the effective residence time of VASP on actin filaments and, therefore, its effective bundling capability. Indeed we observed that the

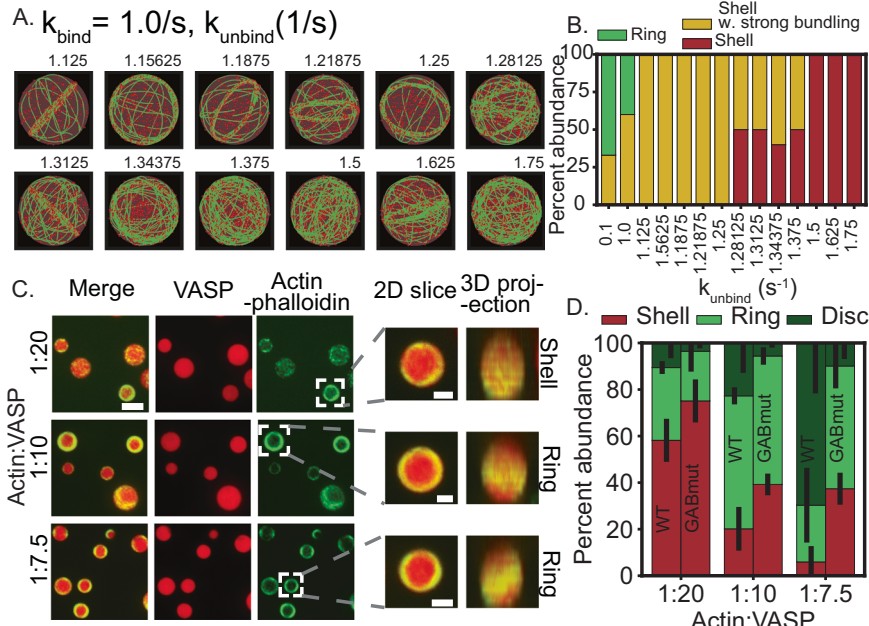

**Fig. 6 | VASP-actin bond lifetime plays a key role in determining the actin network shape. A** Representative final snapshots ($t = 600$) was obtained from simulations for different $k_{unbind}$ values for $k_{bind} = 1.0/s$. **B** The resulting final actin shapes ($N = 10$ for $k_{unbind} = \{1.28125, 1.3125, 1.34375\}$, $N = 5$ otherwise, data from the last 30 snapshots), were used to compute the probability of observing ring-like and shell-like actin. Please refer to Supplementary Fig. 12 for details of PCA and clustering. **C** Experimental images showing VASP-GABmut droplets containing phalloidin-stained actin at various Actin:VASP ratios. Insets show selected droplets with peripheral actin accumulation that were classified as shells or rings from 3D projections. Overview scale bar = 3 μm; Inset scale bar = 1 μm. **D** Experimental data shows mean and standard deviation of percentage abundance corresponding to Shell, Ring, and Disks for WT and GABmut-VASP ($N = 3$ biologically independent experiments). Source data are provided as a Source Data file.

configuration of the actin network changed from rings to shell-ring intermediates (Fig. 6A). Further quantification of the distribution of actin network architectures as $k_{unbind}$ increases shows that weakening bundling by increasing $k_{unbind}$ shifts the abundance of actin networks from rings and shells with strong bundling to shells (Fig. 6B and Supplementary Fig. S12).

In order to test our prediction that a combination of actin-VASP dissociation kinetics determines the network architecture, we sought to mutate VASP in ways that reduce its association with actin filaments. The most obvious way to disrupt actin-VASP interaction would be to mutate VASP's F-actin-binding site. However, this mutation is too severe to be useful in the current study, as it inhibits the bundling of filaments so strongly that peripheral accumulation of actin filaments in VASP droplets is completely abolished[15]. In contrast, mutation of VASP's G-actin-binding (GAB) site has a more modest impact on filament binding and bundling[38,40], such that it provides a useful tool for studying the effects of impaired bundling on the abundance of shells, rings, and disks. To test this idea, we formed droplets of VASP-GABmut[15], as described in Methods, and added unlabeled actin monomers to them. We then stained with phalloidin to specifically visualize actin filaments and performed confocal fluorescence microscopy to determine the spatial arrangement of the filaments within the droplets. Upon three-dimensional reconstruction of the confocal slices, droplets that contained actin shells and rings, along with droplets that deformed into disks, were observed (Fig. 6C). We quantified the abundance of the shells, rings, and disks as a function of actin to VASP ratio (Fig. 6D). Using droplets composed of wild-type VASP, we have previously shown that actin filaments adopt increasingly bundled structures as actin to VASP ratio increases. This increase in bundling results in a higher abundance of rings and disks, and a corresponding decrease in shells (data from ref. 15 replotted). However, using VASP-GABmut, we found that, as the actin to VASP ratio increased, there were substantially fewer droplets that contained bundled actin structures. Specifically, at actin to VASP ratio of 1:7.5, a majority of droplets composed of wild-type

VASP had deformed into disks ($70 \pm 13\%$ mean ± SEM), while less than 10% of droplets composed of VASP-GABmut deformed into disks, with the majority consisting of spherical droplets that contained actin rings ($53 \pm 6\%$ mean ± SEM) (Fig. 6D). Similarly, as the actin to VASP ratio increased for droplets composed of wild-type VASP, fewer contained actin shells, with the majority containing actin rings. In contrast, more than a third of droplets composed of VASP-GABmut contained actin shells, even at an actin to VASP ratio of 1:7.5 ($37 \pm 4\%$ mean ± SEM). Overall, these data suggest that the consequence of reduced filament bundling by VASP-GABmut is a reduction in the number of bundled actin structures, specifically rings and disks, compared to droplets composed of wild-type VASP. Collectively, simulations and experiments confirm that the formation of actin rings requires strong bundling and a large residence time of VASP on actin.

## Actin bundle reorganization determines the maximum aspect ratio of the VASP droplet

Previously, we showed that when actin rings reach a critical thickness, droplets deform and become ellipsoidal[15]. We also showed that at high actin to VASP ratios, the droplet assumes a rod shape, which contains a linear actin bundle (Fig. 5 in ref. 15, reproduced in Fig. 7Ai). Therefore, we next investigated the features of actin rings that determine the aspect ratio of VASP droplets. We begin these simulations with a spherical droplet in which a ring has already formed. To mimic the deformation of the droplet, we change the sphere to a spheroid by changing the major axis and minor axis 10 nm at a time. We then let the system relax into the new droplet geometry for 1 s. In this process, we assume that the relaxation of actin shape results from a combination of mechanical dissipation of excess bending energy and mechanochemical reorganization of crosslinkers. We observe that for a fixed value of maximum filament length ($L_{fil}^{max}$) equaling $\pi R_{drop}$, as we deform the droplet, the actin ring continues to track the surface of the droplet, and at high aspect ratios, the ring unravels to become a linear bundle (Fig. 7Aii and Supplementary Movie M4). Thus, we mimic the

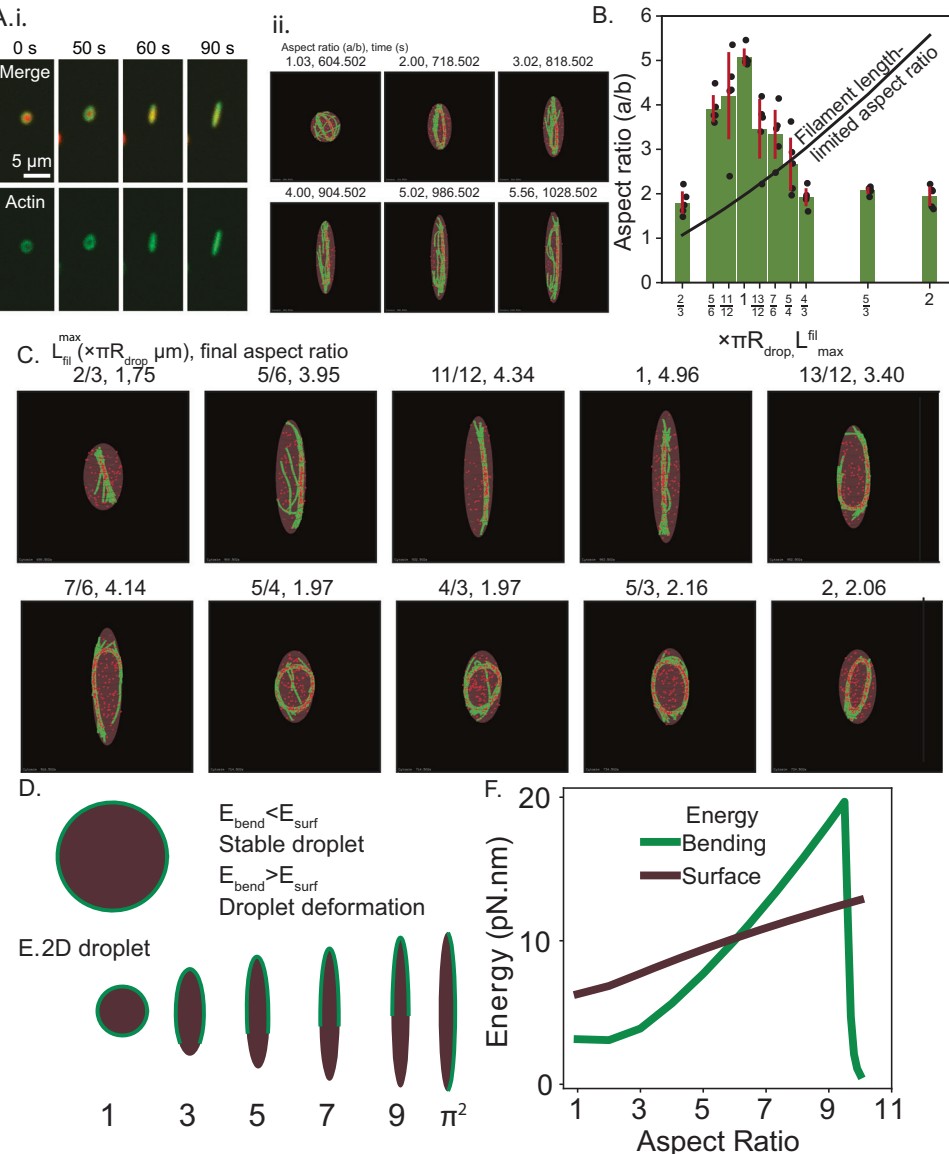

**Fig. 7 | Filament length determines the maximum aspect ratio of the droplet. Ai** Linear droplets that elongate over time are formed by the addition of 2 μM actin to 10 μM VASP droplets Scale bar 5 μm. Reproduced Fig. 5, ref. 15. **Aii.** A time series of snapshots corresponding to $L_{fil}^{max} = \pi R_{drop}$ shows the transition of a network with rings to an ellipsoidal droplet with rings and, eventually, an extended bundle. Please see Supplementary Movie M4 for a video of trajectories. **B** Bar plot of mean and standard deviation in the maximum aspect ratio of droplets at various $L_{fil}^{max}$ values (six replicates). The maximum aspect ratio was determined by the protocol outlined in Supplementary Methods to ensure filaments remain in contact with the high-curvature tips of the ellipsoid. The solid line shows the maximum aspect ratio permitted when the ellipsoidal span, $2a = L_{fil}^{max}$. **C** Representative maximum aspect ratio snapshots from simulations with different $L_{fil}^{max}$. Actin filaments (green) and VASP tetramers (red spheres) are shown within droplets (brown). **D** Energetic considerations causing droplet deformation. **E** Droplet deformation is driven by the competition between actin-bending energy and droplet surface energy. **F** Bending and surface energy as a function of 2d droplet aspect ratio. Source data are provided as a Source Data file.

sphere-to-ellipsoid and ellipsoid-to-rod transitions of the droplet shapes. Next, we varied the maximum filament length and calculated the maximum aspect ratio that was allowed for the droplet (Fig. 7B). The maximum aspect ratio is defined as the droplet shape at which the droplet ends still have adequate actin to remain in contact (Supplementary Fig. 13). The filament length-limited aspect ratio is shown by the solid black line when the major axis of the ellipse equals the maximum final filament length under constant volume constraints (Fig. 7C and Supplementary Fig. 14). We observe that at low maximum filament lengths, the droplets deform to aspect ratios higher than the filament length-limited aspect ratio. This is because the filament bundles are pliant, and filaments can slide past one another to create a bundle longer than the maximum filament length. However, we observe that there is a threshold value of maximum filament length

beyond which the aspect ratio is lower than the filament length-limited aspect ratio. This is because at higher $L_{fil}^{max}$ values, bundles no longer reorganize, and the rings remain kinetically trapped. Thus, we find that while the sphere-ellipse transition occurs under a wider range of ring-forming conditions, the maximum filament length plays a critical role in determining where the ellipse-to-rod transition will take place.

## Discussion

Many actin-binding proteins are thought to undergo liquid-liquid phase separation[90]. The role of these phase-separated domains in organizing actin networks has been studied in different systems. Recent studies have shown that abLIM1 condensates induced asters and webs of F-actin filaments in vitro[4]. Phase separation of Nephrin/Nck/WASP on lipid bilayers increases the dwell time of N-WASP and

Arp2/3 complex, increasing actin assembly[3]. Previously, we showed that the VASP phase separates in liquid-like droplets and that within these droplets, actin can adopt a number of configurations, including shells and rings. We also showed that VASP droplets will deform when actin rings exceed a certain thickness[15].

In this study, we used computational modeling to simulate the interactions between actin and VASP to mimic experimental observations of actin polymerization in a phase-separated VASP droplet. Our simulations revealed that the actin network can become shells, shells with weak bundling, shells with strong bundling, and rings. All these structures form due to stochastic ratcheting of VASP where VASP molecules stabilize the current actin configuration and also steer the network shape changes in the next instance. As a result, the binding of several VASP molecules imposes energetic barriers on the drastic reorganization of actin shape due to kinetic trapping. This ability to overcome barriers is less probable in the experimental timescales. We then focused on the formation of actin rings, which are responsible for droplet deformation. We found that rings that promote deformation of the spherical droplet are made of filaments that are below a certain critical length; above that length, the rings remain trapped in the bundled state and do not straighten out to further deform the droplet. These results may explain the experimental observation that for droplets with the same actin concentration, some deform into rods, but some remain stuck with rings in them (See Fig. 5b of ref. 15). The results presented here are in agreement with other computational models[62,63,72] and are consistent with key experimental observations. Using scaling arguments and simulations, ref. 91 study the interplay between cortical tension and microtubule confinement within platelets and erythroblasts in mammals[91–93]. They show that cortical tension controls the extent and shape of microtubule ring deformation within ellipsoids. To better understand actin organization under confinement, we address the limitations of our model and suggest future directions. We do not consider the complex chemical dynamics of actin filament growth and treadmilling. Ni et al. showed that treadmilling filaments (length smaller than confinement radius) form shells by overcoming myosin-driven contractile forces and reduce bending energy penalty by depositing at the periphery[77]. Additionally, the changes to shell and ring-shaped networks under cytoskeletal components such as myosin motors need to be explored.

The sphere-to-ellipsoid transition and the ellipsoid-to-rod transitions are consistent with our previous energy balance model presented in ref. 15. The first transition from a sphere to a spheroid is consistent with our previous 2D model for droplet deformation, where, circular droplets deform into elliptical droplets when the actin ring bending energy is greater than the droplet surface energy (Fig. 7D). We gain additional insights from the explicit consideration of bundle reorganization. We show that actin networks can form rings over a range of filament lengths. In addition, consistent with experiments, we find that the ellipsoidal droplets can take a wide range of aspect ratios because of the dependence on bundle thickness. For the ellipsoid-to-rod transition, we show that the bundle needs to reorganize by unfolding its curvature, which is consistent with the two energy barriers we identified previously (Fig. 7E, F). Additionally, we see that filament length plays a critical role in aiding the unfolding. Curvature unfolding requires extensive reorganization of the bound VASP crosslinkers to allow for bending energy relaxation of actin filaments. Rings with filaments $L_{fil}^{max}$-$\pi R_{drop}$ deform readily by unfolding, resulting in a bundle longer than $\pi R_{drop}$ and therefore surpassing the filament length-limited aspect ratio. When the filaments are comparable to $\pi R_{drop}$, the stabilization of the ring shape is aided by actin-VASP contacts. Upon droplet shape change, VASP molecules undergo force-sensitive unbinding to dissipate excess mechanochemical energy resulting in an ellipsoid-rod transition. On the other hand, rings made of filaments $L_{fil}^{max} \gg \pi R_{drop}$ cannot reorganize themselves to reach the filament length-limited aspect ratio. Because the filaments forming

such rings trace a significant portion of the ring circumference, the resulting rings resist deformation due to steric-driven restriction on filament unfolding and constant reorganization of actin-VASP contacts. When deformed, such bundles form stable, elliptical rings. Thus, we demonstrate that the internal degrees of freedom of a bundle play a critical role in droplet shape change.

The stabilization of rings with $L_{fil}^{max} \gg \pi R_{drop}$ in the frustrated elliptical shape away from the equilibrium linear bundle can be explained by kinetic trapping. Kinetic trapping stabilizes long-lived steady-states that are far away from the minimum energy configurations[94,95]. Kinetic trapping plays a significant role in self-assembly processes involving multi-body interactions[96–99]. In our study, we see that the rings with $L_{fil}^{max} \gg \pi R_{drop}$ form elliptical rings rather than unfolding to a linear bundle configuration. The stochastic unbinding of each of the four binding sites on VASP tetramers, along with the steric repulsion between actin-actin and actin-VASP molecules, stabilize elliptical configurations. Thus, our results suggest that the relative abundance of rod-shaped droplets should increase at longer time scales. Importantly, kinetic trapping has been observed in various cytoskeletal systems[100,101]. Specifically, crosslinker proteins have been shown to enhance the contractility of actin networks[70] and mitotic spindle assembly[102]. Here, we establish additional mechanisms by which kinetic trapping might guide actin reorganization. It is important to note that we do not directly incorporate the effects of droplet surface tension. Through quasi-static changes to aspect ratio under finite time simulations, we observe bundle configurations that are stabilized by crosslinkers in the bend configuration. It remains to be seen if such structures remain stable under longer time scales as viscous dissipation through crosslinker turnover would lead to internal stress relaxation of the actin network. Additionally, explicit consideration of droplet surface tension through deformable boundaries should be explored in the future.

In summary, our work offers a potentially simple mechanism for the formation of various actin shapes that are relevant to cellular function. Future iterations of such models should include explicit considerations of the viscoelastic nature of the droplet and explicit diffusion of components across the droplet interface. Incorporating such complex physical behaviors would lead to a more robust understanding of the role of such droplets in actin regulation. It would also be insightful to study the formation of protein composites that include multiple actin-binding proteins and their role in determining the actin network architecture. Particularly, additional nucleation in the form of Arp2/3 could help explain the filopodial initiation from lamellipodia as found in cells[103]. As VASP is found close to the membrane, along with a whole host of other actin-binding proteins, it is possible that interactions with other actin-binding proteins could lead to further refinement of the kinetic trapping mechanisms proposed here.

## Methods
### Cytosim simulations
**Chemical and mechanical framework employed in CytoSim.** Simulations were performed in Cytosim (https://gitlab.com/f-nedelec/cytosim). Cytosim is an agent-based framework to simulate the chemical dynamics of filamentous networks while also accounting for their mechanical properties[80,81]. Towards this, Cytosim numerically solves a constrained Langevin framework in a viscous medium at short time intervals (2 ms in this study) to model the dynamics of filaments along with the diffusing species. To capture essential chemical reactions observed in VASP droplets, filament extension is modeled as a deterministic force-sensitive process along with stochastic Monte Carlo sampling of crosslinker binding and its force-sensitive unbinding reactions.

**Representation of actin filaments.** Actin filaments are represented as inextensible fibers whose contour is traced by a series of linear

segments, each of length 100 nm, connected at hinge points (Fig. 1A). CytoSim computes bending energy of the fiber based on the flexural rigidity specified in the input parameters.

**Representation of VASP tetramer.** VASP tetramers are known to bind up to four actin filaments through the F-actin-binding domain[39]. Additionally, VASP also has a G-actin-binding domain which assists in increasing the extension rates of actin filaments[104]. Thus, VASP can act as an actin polymerase and a crosslinking protein[19]. Given that about 60% of amino acids in VASP are disordered[15], we expect VASP to be a flexible crosslinker that can bind over long distances. Therefore, we model VASP as a spherical crosslinker of radius 30 nm with four F-actin-binding sites distributed across the surface of the sphere (Fig. 1B). Cytosim also requires specification of a binding distance parameter between VASP and actin. This binding distance parameter controls F-actin-binding propensity—the reaction propensity is calculated based on the abundance of F-actin found within the sphere centered on the binding site with the radius determined by binding distance. To determine the appropriate binding distance, actin networks were simulated at various crosslinker binding and unbinding rates at binding distances 5, 10, 15, 20, and 30 nm, respectively. We then chose a binding distance of 30 nm because this value allowed us to obtain all the experimentally observed actin networks. VASP molecules can unbind from actin filaments based on unbinding rates specified in the input file. In addition, we also consider Bell's law model to represent slip bond unbinding kinetics.

**Position evolution.** CytoSim employs the Langevin equation to evolve the position of points (both along the fiber and those of VASP tetramers) considered. The discretized points along the filament and the VASP tetramers are considered. For a system of N particles, there are 3N coordinates. For a particle i, the coordinates are given by $x_i = \{x_{i1}, x_{i2}, x_{i3}\}$. The position along each of the dimensions j is evolved according to the stochastic differential equation given by,

$$dx^{ij}(t) = \mu f^{ij}{}_{tot}(t)dt + dB_j(t) \quad (1)$$

Here, μ is the viscosity of solvent, and $f^{ij}{}_{tot}(t)$ represents the total force acting on the particle at time t. The diffusion term (noise) is given by a random variable sampled from a distribution with mean 0 and standard deviation $\sqrt{2D^i dt}$, where the diffusion coefficient is given by $\mu k_B T$ where T is temperature and $k_B$ is Boltzmann constant. Please refer to Supplementary Table 2 for a detailed description of the parameters used in this study.

**Steric considerations.** While the high density increases the propensity of chemical reactions (actin-VASP crosslinking), the steric-driven reduction in mobility within the droplet medium can either increase or decrease the effective rate constant[78]. To capture this effect, we employ a steric repulsion potential between VASP molecules, and actin filaments.

**Actin-covered surface fraction of the droplet.** An icosphere corresponding to the radius of the droplet is generated. The icosphere is subdivided 4x to approximate the curvature of the spherical surface. This was done so the triangle side is comparable to the segmentation length of actin filaments (100 nm). Actin filaments were discretized to the level of monomers and the monomers within 100 nm of the subdivided icosphere were considered to be close to the surface. Each of the monomers were assigned to the icosphere surface triangle closest to it. Thus, every snapshot from the Cytosim simulations was converted to a surface density plot and the fraction of occupied triangles is reported as the actin-covered surface fraction over time.

**Simulations studying the role of droplet deformation.** To understand the role of the actin cytoskeleton in droplet deformation, we build on our previous hypothesis that the droplet deformation is driven by an imbalance between droplet surface energy and actin ring bending energy. Therefore, in this study, we simulate the droplet deformation using the following protocol. The actin filaments are simulated for 600 s under elongation rates and VASP-actin kinetics that favor ring formation. Once the maximum filament length is achieved, the network no longer elongates. Subsequently, we alter the droplet shape iteratively by increasing the Z-axis span of the volume in steps of 10 nm under constant volume conditions. The ring and diffusing VASP tetramers are allowed to equilibrate within the deformed spheroidal boundary shape for 1 s (in Langevin time steps of 2 ms) before undergoing subsequent deformation. The equilibration is achieved by a combination of diffusive dissipation through Langevin dynamics and chemical dissipation from VASP-actin (un)binding, resulting in bundle reorganization. As the largest filament length studied is 2π microns, the highest aspect ratio a/b is achieved when 2a=2πRdrop (when R = 1 mum, (a/b)max ~5.56). The droplets are elongated till we achieve (a/b)max

**Determination of time between two deformations.** To determine the time between two 10 nm deformations, we deformed VASP-only droplets at various rates. At faster rates, we observe that the VASP droplets do not equilibrate by diffusion adequately and are characterized by a region of the droplet without any VASP tetramers. Thus, our droplet deformation approach is a quasi-static approximation of the nonequilibrium process observed in experiments.

**Determining maximum aspect ratio.** Our simulation procedure approximates the droplet deformation process that is driven by competition between actin ring bending forces and the droplet surface tension. Thus, contact of actin along the droplet boundary, particularly the high-curvature caps (ends of the major axis) of the spheroidal droplet is critical. Thus, we need to determine a threshold volume fraction to identify the high-curvature caps and also choose an actin concentration threshold. In this study, we empirically choose a volume fraction of 12.5% and an actin concentration threshold to be $C_{bulk}/3$ as shown in Supplementary Fig. 13. The droplet is deemed to have reached the maximum aspect ratio when the actin concentration within the spheroidal caps falls below the concentration threshold.

**VASP-mutant preparation**

A "cysteine-light" mutant of human VASP was expressed, purified, and labeled as previously described with no modifications[15]. Briefly, pET-6xHis-TEV-KCK-VASP-mutGAB was generated by site-directed mutagenesis to mutate residues RK236, 237EE using the primers GCC AAACTCGAGGAAGTCAGCAAGCAGG and GCTGACTTCCTCGAGTTT GGCTCCAGCAATAG. Gibsol assembly was employed to generate pET-6xHis-TEV-KCK-VASP-mutFAB. The template vector was amplified using the primers CAGCACAACCTTGCCAAGG and AATAGCTGCGGCC AGGCC. The insert fragment encoding the amino acids 225–319 of VASP, with the mutations in the FAB site KR275, 276EE, K278E, and K280E synthesized as gBlock gene fragments from Integrated DNA Technologies with the sequence GGCCTGGCCGCAGCTATTGCTGGA GCCAAACTCAGGAAAGTCAGCAAGCAGGAGGAGGCCTCAGGGGGG CCCACAGCCCCCAAAGCTGAGAGTGGTCGAAGCGGAGGTGGGGGAC TCATGGAAGAGATGAACGCCATGCTGGCCGAGGAAGAGGAAGCCAC GCAAGTTGGGGAGAAAACCCCCAAGGATGAATCTGCCAATCAGGAGG AGCCAGAGGCCAGAGTCCCGGCCCAGAGTGAATCTGTGCGGAGACC CTGGGAGAAGAACAGCACAACCTTGCCAAGG2. VASP droplets were formed by diluting VASP (10% labeled with maleimide AlexaFluor-647) to a final concentration of 20 μM into "droplet buffer": 50 mM Tris pH 7.4, 150 mM NaCl, 5 mM TCEP, 3%(w/v) PEG8k. PEG was added last to induce droplet formation. To visualize the distributions of filamentous

actin within VASP droplets, unlabeled G-actin monomers were added to VASP droplets to achieve varying actin-to-VASP ratios. For all actin to VASP ratios, VASP concentration was kept constant at 20 μM (concentration of VASP monomers, 5 μM VASP tetramers), and the final actin concentration was 1 μM (for an actin:VASP of 1:20), 2 μM (for an actin:VASP of 1:10), and 2.67 μM (for an actin:VASP of 1:7.5). Actin was allowed to polymerize in the VASP droplets for 15 min, then the droplets were stained with Phalloidin-iFluor488 (Abcam) for 15 min to visualize the actin filaments. Confocal z-stacks of droplet images were obtained using an Olympus SpinSR10 spinning disc confocal microscope using a Hamamatsu Orca Flash 4.0V3 sCMOS digital camera. To distinguish the different 3D spatial arrangements of actin filaments within the droplets, 3D projections of the droplets were constructed and analyzed. Specifically, droplets were classified as shells if the 3D reconstruction revealed uniform spherical actin intensity that colocalized with VASP. Rings were defined as droplets with a 3D reconstruction revealing spherical VASP distribution and planar, non-spherical actin distribution. Disks were defined as droplets with a 3D reconstruction in which both VASP and actin colocalized in a non-spherical planar droplet. Image analysis was performed using ImageJ (v.2.1.0).

### Statistics and reproducibility
Simulations discussed in Figs. 2, 3 were generated from three replicates. Five replicates were analyzed to generate data shown in Fig. 4. Ten replicates were used in Fig. 5. Simulations discussed in Fig. 6 consist often replicates for $k_{unbind} = \{1.28125, 1.3125, 1.34375\}$, $N = 5$ otherwise. Experiments with VASP-mutants were repeated independently three times. Five replicates were employed for data shown in Fig. 7. All simulations were initialized with seed filaments with a random distribution of orientation. All trajectory analyses were performed on Python 3.9.

### Reporting summary
Further information on research design is available in the Nature Portfolio Reporting Summary linked to this article.

## Data availability
The LAMMPS input files, initial and final configurations, along with CytoSim input files, are available on GitHub doi:10.5281/zenodo.10714332 [doi.org/10.5281/zenodo.10714332]. Source data for both experiments and simulations are provided with this paper. Source data are provided with this paper.

## Code availability
The Python code used to analyze trajectories will be made available on GitHub https://doi.org/10.5281/zenodo.10714332 [https://doi.org/10.5281/zenodo.10714332]. The source code for CytoSim can be downloaded from GitLab at https://gitlab.com/f-nedelec/cytosim.

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

## Acknowledgements

This work was supported by grants from the National Science Foundation DMS 1934411 to P.R. and J.C.S., Office of Naval Research N00014-20-1-2469 to P.R., National Institutes of Health to P.R. (R01GM132106), and NIH (R35GM139531) to J.C.S. A.C. would like to thank Dr. Yossi Eliaz, Dr. Christopher Lee, Dr. Mayte Bonilla Quintana, Dr. Ashwin Ravichandran and Dr. Sriram Vignesh Mani for constructive feedback on the model and results presented here. They would also like to thank Prof. Francois Nedelec for their feedback on the model schematic.

## Author contributions

A.C., K.G., J.C.S. and P.R. designed experiments. A.C., K.G., J.C.S. and P.R. wrote and edited the manuscript. A.C., K.G., J.C.S. and P.R. performed experiments and analyzed data. All authors consulted on manuscript preparation and editing.

## Competing interests

The authors declare no competing interests.
