## [Peer Review File · Nature Communications]

Reviewers' Comments:

Reviewer #1:

Remarks to the Author:

In the paper titled "Kinetic trapping organizes actin filaments within liquid-like protein droplets", authors Chandrasekaran et al. offer a detailed exploration of the principles that govern actin network organization within liquid-like protein droplets composed of Vasodilator Stimulated Phosphoprotein (VASP), an actin-binding protein. Using a combination of computational modeling and experimental methods, the team provides a nuanced understanding of how VASP-actin interactions create various actin configurations within these droplets, with a key focus on kinetic trapping.

They propose that the formation of different actin configurations such as shells, rings, and mixed states is largely dependent on the kinetics of VASP-actin interactions. Furthermore, the team found that by reducing the VASP's residence time on actin filaments, they could decrease the degree of bundling and thus favor the assembly of shells over rings. The authors corroborated this prediction through experiments with a VASP mutant, which exhibited reduced bundling capabilities.

In their investigation of the transition of droplet shapes, the authors discerned that the transition from spheres to ellipsoids to rods is significantly influenced by ring formation and is dictated by the maximum length of actin filaments. Consequently, the authors identified a set of kinetic parameters that determine the architecture of the actin network and the resultant deformation of VASP droplets.

This study provides valuable insight into the kinetic principles that dictate the structure and dynamics of actin networks within VASP droplets. The remodeling of biological actin has proven to be highly dynamic and complex, largely due to fluctuating developmental and signaling events. Historically, most actin-binding proteins were studied for their individual biochemical activities, primarily in the context of the actin treadmilling process. Yet, intricate signaling necessitates swift reorganization that incorporates simultaneous multiple activities. As demonstrated in this manuscript, multivalent interactions or molecular condensation driven by a single protein could lead to the formation of more complex actin network architectures.

A combination of actin regulatory activities in a spatial, temporally regulated manner is the key to driving the dynamic organization of the actin network. While this manuscript demonstrated a few key interplay between binding, bundling, and filament length, a more comprehensive discussion would help to signify the work further. I have the following comments for the author to elaborate more, which broader biological audiences will more appreciate.

Major comments:

Several highly relevant works on phase-separation regulation in actin reorganization were not discussed in the context of the effect of VASP. These missing studies include those on proteins that only exert a crosslinking effect, such as filamin (as noted by Kimberly L. Weirich et al. in PNAS, 2019), or proteins with both nucleation and crosslinking functions, such as bacterial XopR (Sun H. et al., 2021 Nat Commun) and synaptic PSD condensates (Chen X. et al. eLife, 2023).

The authors did an impressive study in understanding the interplay between elongation rate and bundling and its impact on ring-like structure formation. The transitions from sphere to ellipsoid and then to rod are dependent on filament length, reflecting differences in elongation efficiency between the initial nucleation step and subsequent focused elongation.

During the multi-step process of actin polymerization, proteins like VASP or XopR facilitate the initial formation of short actin filament seeds by nucleation before actively promoting elongation and subsequent bundling. Can the authors delve more into these dynamics of actin remodeling? It would be particularly interesting to discuss this if we consider nucleation as the first step in actin polymerization. This initial step produces filaments that elongate simultaneously before focusing on elongation for treadmilling. Furthermore, it would be beneficial to discuss how changes in the actin:VASP ratio might influence these interplays.

Reviewer #2:

Remarks to the Author:

In this work, Chandrasekaran and coworkers study the behavior of actin crosslinked by tetravalent crosslinkers within spherical confinement, to mimic experiments previously done by these groups where actin and ena/vasp are shown to phase separate into droplets. They also perform new experiments, and study behavior in deformed confinement. As far as what is in the paper, I find the results to be interesting and systematic. However, I am concerned by a lack of comparison to prior work and the correspondence between the confinement model used and the reality of the experiments.

1) If I understand correctly, the experiments from Graham and Stachowiak performed separately and also reported here result in phase separated droplets of Ena/VASP and actin naturally. Although they do not observe actin outside of the droplets, I am not sure that this is well represented by a hard spherical wall. The hard wall completely cuts off density fluctuations on the length scale of the confinement, and even a flexible wall would not be the same restoring effect as the surface tension in experiment, where number fluctuations of the components are controlled by the chemical potentials inside and outside and there is only an emergent effective confining force.

I presume the authors were not able to assemble phase separated droplets using this combination of actin and multivalent crosslinkers. Could they speculate more about why that is and the differences that could arise there and the current model here?

2) Even if the authors do want to stick to spherical confinement, as far as I can tell there is only one passing reference to a prior work on this topic (Ref 46), however, there is much more work that has been done on confined actin and crosslinkers, so I feel this work should be put in the context of these prior works. Not having read all of those papers recently, I cannot exactly say how many conclusions here have already been determined if any, but this is a serious oversight that must be corrected.

For example:

<https://www.nature.com/articles/s42003-021-02653-6>

<https://www.frontiersin.org/articles/10.3389/fmolb.2020.610277/full>

https://pubs.rsc.org/en/content/articlehtml/2011/sm/c1sm06060k?casa_token=_nu_2pPI1V0AAA-AA:3d7Auuo2mj6fq7cNXwj2ONzCeX-2asYa__PzDRTwztc8UjI3R_-CG43X7Dj6EtPIJ_VetmE63n2PEY0G

[https://www.cell.com/biophysj/pdf/S0006-3495\(22\)03862-0.pdf](https://www.cell.com/biophysj/pdf/S0006-3495(22)03862-0.pdf)

<https://onlinelibrary.wiley.com/doi/full/10.1002/cm.21565>

(and also work on Microtubules, e.g. from Nedelec, e.g

<https://www.sciencedirect.com/science/article/pii/S0960982209010252>)

Within the modeling, I have two questions:

- 1) What if anything did the authors modify in Cytosim? Are the tetravalent crosslinkers already in existence or is this a new development.
- 2) I'm not sure what the authors mean by a force sensitive unbinding reaction (and the details are missing in the simulation methods). Do they mean that slip bonding/bell's law kinetics are used (as described here <https://gitlab.com/f-nedelec/cytosim/-/blob/master/doc/sim/stochastic.md>)?

My belief from prior papers and work on this topic is that cytosim does not (attempt to) satisfy detailed balance (as is stated on the bottom of page 4 in this paper), and also that pure slip bonding kinetics is inconsistent with detailed balance since the ratio of on and off rates depends on force and not energy.

Lastly, methods are not yet available here:

https://github.com/achansek/VADroplet_CytoSim_analyses

Response to the reviewer comments on the manuscript titled “Kinetic trapping organizes actin filaments within liquid-like protein droplets”

We thank the reviewers for offering valuable feedback that has helped us improve the manuscript and address any points of confusion. We present point-by-point responses to reviewer comments below. We have changed the manuscript and Supplementary Material (highlighted in blue).

Reviewer #1 (Remarks to the Author):

Reviewer summary

In the paper titled "Kinetic trapping organizes actin filaments within liquid-like protein droplets", authors Chandrasekaran et al. offer a detailed exploration of the principles that govern actin network organization within liquid-like protein droplets composed of Vasodilator Stimulated Phosphoprotein (VASP), an actin-binding protein. Using a combination of computational modeling and experimental methods, the team provides a nuanced understanding of how VASP-actin interactions create various actin configurations within these droplets, with a key focus on kinetic trapping.

They propose that the formation of different actin configurations such as shells, rings, and mixed states is largely dependent on the kinetics of VASP-actin interactions. Furthermore, the team found that by reducing the VASP's residence time on actin filaments, they could decrease the degree of bundling and thus favor the assembly of shells over rings. The authors corroborated this prediction through experiments with a VASP mutant, which exhibited reduced bundling capabilities.

In their investigation of the transition of droplet shapes, the authors discerned that the transition from spheres to ellipsoids to rods is significantly influenced by ring formation and is dictated by the maximum length of actin filaments. Consequently, the authors identified a set of kinetic parameters that determine the architecture of the actin network and the resultant deformation of VASP droplets.

This study provides valuable insight into the kinetic principles that dictate the structure and dynamics of actin networks within VASP droplets. The remodeling of biological actin has proven to be highly dynamic and complex, largely due to fluctuating developmental and signaling events. Historically, most actin-binding proteins were studied for their individual biochemical activities, primarily in the context of the actin treadmilling process. Yet, intricate signaling necessitates swift reorganization that incorporates simultaneous multiple activities. As demonstrated in this manuscript, multivalent interactions or molecular condensation driven by a single protein could lead to the formation of more complex actin network architectures.

A combination of actin regulatory activities in a spatial, temporally regulated manner is the key to driving the dynamic organization of the actin network. While this manuscript demonstrated a

few key interplay between binding, bundling, and filament length, a more comprehensive discussion would help to signify the work further. I have the following comments for the author to elaborate more, which broader biological audiences will more appreciate.

Response: We thank the reviewer for their thoughtful, positive, and extensive feedback. We appreciate that the reviewer agrees with us on the importance of our findings and their implications in actin network formation. As discussed below, we have extensively modified the revised manuscript to address the points raised by the reviewer.

Major comments:

1.1 Several highly relevant works on phase-separation regulation in actin reorganization were not discussed in the context of the effect of VASP. These missing studies include those on proteins that only exert a crosslinking effect, such as filamin (as noted by Kimberly L. Weirich et al. in PNAS, 2019), or proteins with both nucleation and crosslinking functions, such as bacterial XopR (Sun H. et al., 2021 Nat Commun) and synaptic PSD condensates (Chen X. et al. eLife, 2023).

Response: We thank the reviewer for pointing out an opportunity to expand the scope of our work to be more complete. We have now expanded the literature review in our introduction and discussion to reflect these suggestions and included all the references. We hope the reviewer will agree that the revised version of the text is significantly more comprehensive.

Action taken: Extensive revisions to the introduction and discussion along with an expanded literature survey.

1.2 The authors did an impressive study in understanding the interplay between elongation rate and bundling and its impact on ring-like structure formation. The transitions from sphere to ellipsoid and then to rod are dependent on filament length, reflecting differences in elongation efficiency between the initial nucleation step and subsequent focused elongation.

During the multi-step process of actin polymerization, proteins like VASP or XopR facilitate the initial formation of short actin filament seeds by nucleation before actively promoting elongation and subsequent bundling. Can the authors delve more into these dynamics of actin remodeling? It would be particularly interesting to discuss this if we consider nucleation as the first step in actin polymerization. This initial step produces filaments that elongate simultaneously before focusing on elongation for treadmilling.

Response: We thank the reviewer for this suggestion. We have now conducted additional simulations to decipher the impact of actin nucleation on bundling. As shown in Figure S10 (reproduced below), we find that when the nucleation rate is very fast, the filament length decreases and actin accumulates in the center of the droplet.

Action taken: We conducted additional simulations as described above to investigate the role of actin nucleation. These results are now presented in Supplementary Figure S10, reproduced below along with the caption for ease of review.

Figure S10: Broadly, we show that excess nucleation causes actin accumulation closer to the droplet center. This is explained by changes to the filament length distribution. We observe that nucleation for a given maximum F-actin concentration reduces the median filament length of actin filaments. At low nucleator concentrations, we find that the long filaments ($\geq 2R_{\text{drop}}$) form actin rings while the short filaments ($< 2R_{\text{drop}}$) are crosslinked to the long filaments resulting in a diffuse ring. Above a threshold concentration of the nucleator, the population of long filaments reduces drastically and VASP-crosslinking bundles these short filaments resulting in actin accumulation in the droplet core rather than the periphery.

Figure S10. Nucleation causes changes to filament length distribution causing actin accumulation in the droplet core. A. Representative final snapshots ($t=600$) from simulations at various linear nucleator concentrations. Actin filaments are shown in green while VASP tetramers are shown as red spheres. B. Time series of the number of filaments in the system at various nucleator concentrations. The mean is shown as solid line while standard deviation is shown as shaded area (5 replicates). C. The distribution of filament lengths at various nucleator concentrations are plotted as cumulative density functions. The median density is represented

by the dotted line. D. Stacked bar graph shows the fraction of VASP tetramers that are free and bound to 1, 2, 3, and 4 filaments. Corresponding nucleator concentration is shown in X-axis. E. Mean surface area covered by actin is shown as a bar graph at various nucleator concentrations. Standard deviation is shown in red error bars. F. The droplet is divided into concentric shells of radius r , and thickness $\delta r = 25 \text{ nm}$. The ratio of density of actin within shells and the bulk density ($\rho(R_{drop})$) is plotted at various nucleator concentrations. B., C, D., and E..
Data used: Last 30 snapshots from each of the 5 replicates per nucleator concentration. F. Data used: Last 10 snapshots from each of the 5 replicates per nucleator concentration.

1.3 Furthermore, it would be beneficial to discuss how changes in the actin:VASP ratio might influence these interplays.

Response: We thank the reviewer for this suggestion. As described below, we have conducted additional simulations to study the role of actin:VASP ratio and included these results in the revised manuscript.

Action taken: We conducted additional simulations as described above to investigate the role of actin:VASP ratio. These results are now presented in Figure S11, reproduced below along with the caption for ease of review.

In our original submission, we studied different concentrations of VASP-tet and showed that ring formation was robust in the concentration ranges studied. Here, we increase actin concentration within the droplet by starting with different numbers of seed filaments that grow at $0.0103 \mu\text{m}/\text{s}$. We find that at any given VASP concentration, 30 filaments can be effectively crosslinked to form rings. As the number of filaments increases, VASP crosslinking fails to adequately crosslink the entire network into rings. As a result, we find a new network organization where sub-sections of the network are bundled while the rest of the network remains diffuse. Additional experiments are needed to see if such high actin concentrations can be sustained stably within a VASP droplet.

Figure S11. Networks with high actin concentration are characterized by diminished actin bundling. A. Representative final snapshots (t=600s) from simulations at various number of filaments and [VASP-tet]. Filaments are shown in green while VASP tetramers are shown as red spheres. B. Feature and cluster optimization to determine salient network shapes. Dataset from rings, and rings with strong bundling from Figure 2 were combined with data from simulations in panel A (6 replicates). i. Two principal components explain 96.86% of variance in original data. ii. Varimax loading shows that PCs 1, 2, and 3 take information from fraction of VASP bound to {1,2,4} filaments, Actin-covered surface area and VASP bound to 3 filaments respectively. iii. Silhouette coefficient suggests using K-means clustering (with two PCs) shows the network has three clusters. C. Plot of first two PCs, where spheres represent data points corresponding to rings and rings with strong bundling. Squares represent data from the last 30 snapshots (5%) from each replicate shown in panel A. Data points are colored by cluster. Snapshot from dataset in panel A. closest to the centroid of each cluster is shown along with ([VASP-tet], Number of filaments). D. Probability of various network shapes is shown at various Number of filaments and [VASP-tet].

Reviewer #2 (Remarks to the Author):

Reviewer Summary

In this work, Chandrasekaran and coworkers study the behavior of actin crosslinked by tetravalent crosslinkers within spherical confinement, to mimic experiments previously done by these groups where actin and ena/vasp are shown to phase separate into droplets. They also perform new experiments, and study behavior in deformed confinement. As far as what is in the paper, I find the results to be interesting and systematic. However, I am concerned by a lack of comparison to prior work and the correspondence between the confinement model used and the reality of the experiments.

Response: We thank the reviewer for their careful and thoughtful assessment of our work and for recognizing the impact of our work. We understand the concern about the confinement and comparison with the model and we elaborate the steps we have taken to address this concern.

2.1) If I understand correctly, the experiments from Graham and Stachowiak performed separately and also reported here result in phase separated droplets of Ena/VASP and actin naturally. Although they do not observe actin outside of the droplets, I am not sure that this is well represented by a hard spherical wall. The hard wall completely cuts off density fluctuations on the length scale of the confinement, and even a flexible wall would not be the same restoring effect as the surface tension in experiment, where number fluctuations of the components are controlled by the chemical potentials inside and outside and there is only an emergent effective confining force. I presume the authors were not able to assemble phase separated droplets using this combination of actin and multivalent crosslinkers. Could they speculate more about why that is and the differences that could arise there and the current model here?

Response: The reviewer is correct in pointing out that we did not consider the fluctuations of VASP and actin at the interface. To address this, in the current submission, we explore detailed molecular dynamics simulations in LAMMPS of VASP droplets where surface tension is emergently incorporated. Inspired by LAMMPS simulations of nuclear condensates by Cho et al.¹ and actin bundling by Wang et al.², we develop a VASP-actin droplet system in this submission. We consider demixing of VASP molecules from crowder molecules (approximates role of PEG in experiments) by explicitly considering both molecules as spheres with appropriate Lennard-Jones interactions as explained in Supplementary Methods. We then add actin filaments and allow the filaments to be cross-linked. We show that VASP forms a droplet (Figure S1A, B) and that actin filaments remain confined inside the droplet (Figure S1C, D). Additionally, we also show that the actin filaments cause droplet deformation (Figure S1 E, F). We believe this is a valuable area to explore in our research and have mentioned it in the list of future directions (Discussion section).

We also acknowledge that the flexible boundary protocol is not equivalent to the surface tension of the droplet. In this study, we concentrate on the actin organization and have explored how actin filament length could drive the deformation. We have elaborated on this limitation in our discussion section.

We are unsure of what chemical potential differences the reviewer is referring to because, in these phase-separated droplets, chemical potential of the species across the interface is the same under equilibrium conditions (Figure 4 in Hyman Annual reviews³). We acknowledge that despite this simplification, our model is able to capture the main experimental effects. The LAMMPS simulations are not currently computationally scalable to study the role of critical biophysical parameters. Additionally, we would need to extensively optimize this simulation framework and related parameters to generate quantitative comparisons with the experiments. Therefore, we chose to approximate the boundary as a confining hard wall boundary and studied the system using CytoSim.

Action taken: Text added to discussion to state that the role of fluctuations must be investigated carefully in future studies and new supplementary Figure S1 showing the detailed simulation of actin enclosed VASP droplets using LAMMPS. Figure is reproduced with caption for ease of review.

Figure S1. Detailed model for actin encapsulated droplet qualitatively matches continuum model predictions. A. Droplet mimic consisting of VASP molecules (LJ), Crowder (visualized at a smaller particle size, colored gray) and actin filaments (shown in panels C and D as green filaments) are generated in LAMMPS. Please refer to Supplementary Methods and Table S1 for detailed description of method and the parameters used. B. Radial density distribution of VASP (Droplet) and Crowder molecules. C. Final snapshot of droplet containing six actin filaments (14×10^7 steps). Right subpanel shows corresponding actin configuration. Crosslinks between actin filaments are visualized as purple bonds. D. Final snapshot of droplet containing twenty actin filaments (14×10^7 steps). Right subpanel shows corresponding actin configuration. Crosslinks between actin filaments are visualized as purple bonds. E. Final asphericity index and F. Aspect ratio of droplet phase are plotted. Droplet interface was determined as the distance along radial density profile where density falls below 0.1.

1. Cho, E. J. & Kim, J. S. Crowding effects on the formation and maintenance of nuclear bodies: insights from molecular-dynamics simulations of simple spherical model particles. *Biophys. J.* **103**, 424–433 (2012).
2. Wang, Y. & Qian, J. Buckling of filamentous actin bundles in filopodial protrusions. *Acta Mech. Sin.* **35**, 365–375 (2019).
3. Jülicher, F. & Weber, C. A. Droplet Physics and Intracellular Phase Separation. (2023) doi:10.1146/annurev-conmatphys-031720-032917.

2.2 Even if the authors do want to stick to spherical confinement, as far as I can tell there is only one passing reference to a prior work on this topic (Ref 46), however, there is much more work that has been done on confined actin and crosslinkers, so I feel this work should be put in the context of these prior works. Not having read all of those papers recently, I cannot exactly say how many conclusions here have already been determined if any, but this is a serious oversight that must be corrected.

<https://www.nature.com/articles/s42003-021-02653-6>

<https://www.frontiersin.org/articles/10.3389/fmolb.2020.610277/full>

https://pubs.rsc.org/en/content/articlehtml/2011/sm/c1sm06060k?casa_token=_nu_2pPI1V0AAAAA:3d7Auuo2mj6fq7cNXwj20NzCeX-2asYa__PzDRTwztc8UjI3R_-CG43X7Dj6EtPIJ_VetmE63n2PEY0G

[https://www.cell.com/biophysj/pdf/S0006-3495\(22\)03862-0.pdf](https://www.cell.com/biophysj/pdf/S0006-3495(22)03862-0.pdf)

<https://onlinelibrary.wiley.com/doi/full/10.1002/cm.21565>

(and also work on Microtubules, e.g. from Nedelec, e.g. <https://www.sciencedirect.com/science/article/pii/S0960982209010252>)

Response: We thank the reviewer for pointing this out. As noted in the letter to the editor, we have now extensively revised the manuscript to address this point. We hope the reviewer will agree that this version of the manuscript is more comprehensive in placing our findings in the context of the literature. Additionally, we note that in the LAMMPS simulation figure in Figure S1 D,E, we observe departures from spherical geometry, reinforcing the slow deformation analysis in Figure 7 of the manuscript.

Action taken: extensive edits to the text to include relevant references and place the results in context. Deformation from spherical geometry was observed in the LAMMPS simulations (Figure S1).

For example:

2.3 Within the modeling, I have two questions:

2.3.1) What if anything did the authors modify in Cytosim? Are the tetravalent crosslinkers already in existence or is this a new development.

Response: We did not make any modifications to Cytosim in this work. The tetravalent crosslinkers were already available in the code base and used in this paper⁴

4. Eliaz, Y., Nedelec, F., Morrison, G., Levine, H. & Cheung, M. S. Insights from graph theory on the morphologies of actomyosin networks with multilinkers. *Phys Rev E* **102**, 062420 (2020).

Action taken: We have explicitly cited this paper in the current submission.

2.3.2) I'm not sure what the authors mean by a force sensitive unbinding reaction (and the details are missing in the simulation methods). Do they mean that slip bonding/bell's law kinetics are used (as described here <https://gitlab.com/f-nedelec/cytosim/-/blob/master/doc/sim/stochastic.md>)?

Response: Thank you for clarifying. Indeed we use the slip bond, and for consistency, the text has been modified to Bell's law throughout.

Action taken: text modification for consistency.

2.4 My belief from prior papers and work on this topic is that cytosim does not (attempt to) satisfy detailed balance (as is stated on the bottom of page 4 in this paper), and also

that pure slip bonding kinetics is inconsistent with detailed balance since the ratio of on and off rates depends on force and not energy.

Response: We apologize for the miscommunication and have clarified the text in our current submission. We have eliminated the mention of detailed balance in the revised manuscript.

Action taken: text modifications for consistency.

2.5) Lastly, methods are not yet available here:

https://github.com/achansek/VAdroplet_CytoSim_analyses

Response: we have now provided the input files, analysis scripts, along with the simulation analysis data used to plot Figures in the following GitHub link https://github.com/RangamaniLabUCSD/VASP_droplet_CytoSim/tree/main

Action taken: link updated.

Reviewers' Comments:

Reviewer #1:

Remarks to the Author:

The authors have provided a commendable response to my previous comments, marked by thoroughness and a meticulous approach to the feedback received. The study's exploration of dynamic actin remodeling across different cellular environments is crucial. It aptly investigates how various kinetic parameters shape the emergence of unique actin network architectures alongside actin-binding proteins or within droplets. The focus on the actin to VASP ratio is particularly laudable, as it greatly enhances the value of the research.

In summary, the authors have effectively addressed the comments. Their diligent expansion of the literature review and the inclusion of additional simulations, presented as two supplementary figures, provide a well-rounded and detailed discussion on the interplay between nucleation and crosslinking and the impact of varying stoichiometry between actin and VASP. Given the innovative and high-quality nature of this research, I recommend its publication in Nature Communications.

Reviewer #2:

Remarks to the Author:

The authors have made a good faith attempt to address all comments.

Also, to clarify the point about chemical potentials, if the system were at equilibrium (as opposed to non-equilibrium) then eventually the chemical potentials would be equal. However, the variance in particle number of each species still depends on the chemical potential:

$$\beta \var N_i = (d \mu_i / dN_i)^{-1}$$

REVIEWERS' COMMENTS

Reviewer #1 (Remarks to the Author):

The authors have provided a commendable response to my previous comments, marked by thoroughness and a meticulous approach to the feedback received. The study's exploration of dynamic actin remodeling across different cellular environments is crucial. It aptly investigates how various kinetic parameters shape the emergence of unique actin network architectures alongside actin-binding proteins or within droplets. The focus on the actin to VASP ratio is particularly laudable, as it greatly enhances the value of the research.

In summary, the authors have effectively addressed the comments. Their diligent expansion of the literature review and the inclusion of additional simulations, presented as two supplementary figures, provide a well-rounded and detailed discussion on the interplay between nucleation and crosslinking and the impact of varying stoichiometry between actin and VASP. Given the innovative and high-quality nature of this research, I recommend its publication in Nature Communications.

We thank the reviewer for the positive comments on our revised manuscript.

Reviewer #2 (Remarks to the Author):

The authors have made a good faith attempt to address all comments.

Also, to clarify the point about chemical potentials, if the system were at equilibrium (as opposed to non-equilibrium) then eventually the chemical potentials would be equal. However, the variance in particle number of each species still depends on the chemical potential:

$$\beta \var N_i = (d \mu_i / dN_i)^{-1}$$

We thank the reviewers for their positive feedback that helped improve this paper. We also thank the reviewer for pointing out the relationship copy number fluctuation and chemical potential. This is an additional feature to explore in our subsequent simulations.